# *KCNK18* Biallelic Variants Associated with Intellectual Disability and Neurodevelopmental Disorders Alter TRESK Channel Activity

**DOI:** 10.3390/ijms22116064

**Published:** 2021-06-04

**Authors:** Lisa Pavinato, Ehsan Nematian-Ardestani, Andrea Zonta, Silvia De Rubeis, Joseph Buxbaum, Cecilia Mancini, Alessandro Bruselles, Marco Tartaglia, Mauro Pessia, Stephen J. Tucker, Maria Cristina D’Adamo, Alfredo Brusco

**Affiliations:** 1Department of Medical Sciences, University of Turin, 10126 Turin, Italy; lisa.pavinato@unito.it; 2Center for Molecular Medicine Cologne, Institute of Human Genetics, University of Cologne, 50931 Cologne, Germany; 3Department of Physiology and Biochemistry, Faculty of Medicine and Surgery, University of Malta, MSD-2080 Msida, Malta; ehsan.nematian@um.edu.mt (E.N.-A.); mauro.pessia@um.edu.mt (M.P.); 4Unit of Medical Genetics, “Città della Salute e della Scienza” University Hospital, 10126 Turin, Italy; andrea.zonta@cittadellasalute.to.it; 5Seaver Autism Center for Research and Treatment, Icahn School of Medicine at Mount Sinai, New York, NY 10029, USA; silvia.derubeis@mssm.edu (S.D.R.); joseph.buxbaum@mssm.edu (J.B.); 6Department of Psychiatry, Icahn School of Medicine at Mount Sinai, New York, NY 10029, USA; 7The Mindich Child Health and Development Institute, Icahn School of Medicine at Mount Sinai, New York, NY 10029, USA; 8Friedman Brain Institute, Icahn School of Medicine at Mount Sinai, New York, NY 10029, USA; 9Department of Genetics and Genomic Sciences, Icahn School of Medicine at Mount Sinai, New York, NY 10029, USA; 10Department of Neuroscience, Icahn School of Medicine at Mount Sinai, New York, NY 10029, USA; 11Genetics and Rare Diseases Research Division, Ospedale Pediatrico Bambino Gesù, IRCCS, 00165 Rome, Italy; cecilia.mancini@opbg.net (C.M.); marco.tartaglia@opbg.net (M.T.); 12Department of Oncology and Molecular Medicine, Istituto Superiore di Sanità, 00161 Rome, Italy; alessandro.bruselles@iss.it; 13Department of Physiology, College of Medicine and Health Sciences, United Arab Emirates University, Al Ain P.O. Box 17666, United Arab Emirates; 14Clarendon Laboratory, Department of Physics, University of Oxford, Oxford OX1 4BH, UK; stephen.tucker@physics.ox.ac.uk

**Keywords:** *KCNK18*, TRESK, K2P, intellectual disability, potassium channel, autism spectrum disorder, ASD

## Abstract

The TWIK-related spinal cord potassium channel (TRESK) is encoded by *KCNK18*, and variants in this gene have previously been associated with susceptibility to familial migraine with aura (MIM #613656). A single amino acid substitution in the same protein, p.Trp101Arg, has also been associated with intellectual disability (ID), opening the possibility that variants in this gene might be involved in different disorders. Here, we report the identification of *KCNK18* biallelic missense variants (p.Tyr163Asp and p.Ser252Leu) in a family characterized by three siblings affected by mild-to-moderate ID, autism spectrum disorder (ASD) and other neurodevelopment-related features. Functional characterization of the variants alone or in combination showed impaired channel activity. Interestingly, Ser252 is an important regulatory site of TRESK, suggesting that alteration of this residue could lead to additive downstream effects. The functional relevance of these mutations and the observed co-segregation in all the affected members of the family expand the clinical variability associated with altered TRESK function and provide further insight into the relationship between altered function of this ion channel and human disease.

## 1. Introduction

*KCNK18* encodes the TWIK-Related Spinal cord K^+^ channel (TRESK), a member of the two-pore domain (K2P) potassium channel family. K2P channels count 15 members, encoded by different *KCNK* genes [1]. At the protein level, these channels have different names according to their structural and functional similarities. They are classified into the TWIK, TREK, TASK, TALK, THIK and TRESK subfamilies [2].

The molecular architecture of the TRESK channel resembles other K2P channel subtypes except that it has a longer intracellular loop between the second and the third transmembrane domains and a shorter C-terminal tail. The intracellular loop contains fundamental modulatory sites such as phosphorylation, dephosphorylation and calcineurin binding motifs. Under resting conditions, murine TRESK is phosphorylated and maintains an inactive conformation, but, after a calcium signal, the channel is rapidly activated following dephosphorylation by the calcium-dependent phosphatase calcineurin [3,4]. To allow this process, calcineurin must bind to the channel through interaction with its non-catalytic sites, helped by the regulatory protein, 14-3-3. In humans, the process is also supported by protein kinase C (PKC) [3]. The switch between the active and resting states has been intensely investigated [4] and it is mediated by the phosphorylation of TRESK at two different regulatory regions, the 14-3-3 binding site (Serine 252, Ser252, hereafter) and three adjacent serine residues (Ser262, Ser264 and Ser267, collectively named serine cluster, hereafter). These residues are conserved in murine TRESK as Ser264, Ser274, Ser276 and Ser279. 

The activated calmodulin complex stimulates the binding of calcineurin to the TRESK intracellular loop, thus promoting dephosphorylation of the two regulatory residues (Ser252 and the serine cluster) and the activation of the channel. After the decay of the calcium signal, channel activity is inhibited by the restoring of the phosphorylated state by protein kinase A, which acts at the level of Ser252, and microtubule affinity-regulating (MARK) kinases, which mediate phosphorylation at the level of the serine cluster, supported by 14-3-3 proteins [4].

Interestingly, a high basal activity and reduced response to calcium have been reported in a study using a murine TRESK mutant characterized by a constitutively dephosphorylated state of the serine cluster [4]. Remarkably, prolonged phosphorylation of this motif has been shown to cause reduced responsiveness of the channel and low basal activity [4,5], further documenting the complex modulatory process controlling TRESK function.

*KCNK18* expression in the dorsal root (DRG) and trigeminal (TG) ganglia [6] led to a proposed role for this channel in a variety of pain pathways [7]; moreover, an involvement in typical migraine with aura has been suggested by different authors [8,9,10,11]. Despite the expression of *KCNK18* in the basal ganglia, its mRNA is detectable also in other brain compartments, including the hypothalamus, frontal cortex, cortex, anterior cingulate cortex, hippocampus, spinal cord and substantia nigra (www.gtexportal.org (accessed on 22 April 2021)), suggesting a wider involvement of the channel in physiological brain function. Interestingly, recent studies have also reported *KCNK18* variants in patients with developmental delay (DD) and intellectual disability (ID), with migraine as an associated feature in some cases [12,13]. Here, we describe the identification of a family with three siblings affected by mild to moderate ID, seizures, and autistic-like behavior with different degrees of severity, carrying biallelic *KCNK18* variants, supporting a possible involvement of the gene in neurodevelopmental disorders (NDD).

## 2. Results

### 2.1. Clinical Features of the Three Affected Siblings

A family of European origins, with three siblings affected by an unclassified NDD, was studied (Figure 1A). The parents had no history of NDD or of any notable diseases, except for a carcinoma in the mother. The first proband was a 26-year-old female at the time of the last evaluation. Born at term after cesarean delivery, the proband showed a delay in the acquisition of developmental milestones, with the first phonemes at 18 months, walking at 24 months and sphincter control at 5 years. Mild ID was reported (total IQ—64, verbal IQ—60, performance IQ—70), with difficulties in calculation and inability to recognize the value of money. Poor language, dysarthria and attention deficit were observed. Moreover, partial temporal–spatial disorientation, poor criticism, and judgment with limitation of the possibilities of self-determination were reported. The support from a caretaker was required for everyday activity and for self-care. She suffered frequent (once a week) frontal cephalalgy. At clinical examination, the behaviour was uncooperative and restless. Even if autism spectrum disorder was not formally assessed using the Autism Diagnostic Observation Schedule (ADOS-2), nor using other behavioural tests, the patient showed an important impairment of relationships that, combined with the other symptoms, led the physician to include her in the autism spectrum.

The second born proband was a 19-year-old female diagnosed with a mild ID (total IQ—56, verbal IQ—47, performance IQ—76). The patient showed compromised social relations that, as with her sister, included the patient into a clinical diagnosis of autism spectrum disorder. Specific difficulties in the logical-mathematical area were reported, together with the inability to remember new concepts. The support from a second person was necessary for everyday activity. An episode of epileptic seizure was reported and supported by electroencephalography (EEG), which showed comitiality. 

The last born was a 16-year-old male affected by dyslexia and difficulties in learning, but no further information were available.

The social-economic situation of the family was also taken into account, to understand if the impaired behaviour of the patients could be caused by a combination of genetic and environmental factors [14,15]; no noteworthy situation was identified, except for a worsening of the already poor social behaviour of the patients after the death of their mother.

### 2.2. Genomic and Structural Analyses

The patients were enrolled in the Autism Sequencing Consortium (ASC) project [16], and a WES of the whole family members performed. In the three affected members, the analysis led to the identification of compound heterozygosity for two missense changes (c.487T > G, p.Tyr163Asp, paternally inherited; c.755C > T, p.Ser252Leu, maternally inherited) (Figure 1A). Both variants affected residues that are highly conserved among orthologs (Figure 1B) and were predicted to have a disruptive impact on protein function (Appendix A). Variant p.(Tyr163Asp) had not previously been reported in GnomAD database (ver 2.1.1) (www.gnomad.broadinstitute.org (accessed on 22 April 2021)), while variant p.(Ser252Leu) had an allelic frequency of 0.0001379 at the time of the analysis; both the variants were not previously reported in our in-house databases. Both affected residues are located within the intracellular domains of the channel (Figure 1C), a region that plays a key role in the conformational switching events controlling TRESK function [5]. Of note, Ser252 plays an important regulatory role in the switching process by cycling between its phosphorylated and dephosphorylated states [5]. The highly conserved nature of the two residues and their predicted location consistently supported the functional relevance of the two amino acid substitutions. WES data analysis excluded the occurrence of other functionally relevant variants compatible with known Mendelian disorders based on the expected inheritance model and clinical presentation, further supporting the relevance of the two missense variants involving KCNK18.

### 2.3. p.Tyr163Asp and p.Ser252Leu Do Not Alter Basal TRESK Function Genomic and Structural Analyses

To validate the *in silico* predictions, we next examined whether the two missense changes in *KCNK18* affect the functional properties of TRESK. Figure 2A shows representative whole-cell basal currents recorded from *Xenopus* oocytes injected with mRNA encoding human wild-type (WT) or the mutant TRESK channels. All channel types exhibited similar outwardly rectifying K^+^ current amplitudes, but the current in oocytes expressing TRESK^S252L^ was slightly larger (Figure 2A,B). This larger basal K^+^ current, which is not statistically different, could be due to a reduction in the phosphorylation state of the mutant channel compared to TRESK^WT^. However, a robust gain of function in the basal K^+^ current has been observed when a serine close to the cluster was replaced with an alanine. Given that the K2P channels work as dimers, to mimic the compound heterozygous condition of affected subjects, equal amounts of the *KCNK18* mutated mRNAs were co-injected into oocytes at a 1:1 ratio. Co-injection of mutant mRNAs did not substantially affect the whole-cell basal currents, which had similar amplitudes to those reported in oocytes overexpressing WT TRESK (Figure 2A,B). TRESK channels typically produce a leak K^+^ current that stabilizes the resting membrane potential [13]. We observed that the expression of WT TRESK or both mutants, expressed separately or together, shifted the resting cell membrane potential of the oocytes towards the K^+^ equilibrium potential (E_K_) (Figure 2C). Assuming an intracellular K^+^ concentration of approximately 140 mM in the oocyte, the predicted value for E_K_ is −107 mV. A similar hyperpolarizing effect was observed upon expression of all TRESK channel types (Figure 2C). Collectively, these findings indicate that both mutations do not dramatically change the basal activity of TRESK channels, which also retain their ability to control the cell’s resting potential. 

### 2.4. p.Tyr163Asp and p.Ser252Leu Impair the Ability of TRESK to Properly Respond to Calcineurin Activation

Given that the variants lie in amino acids located in regions important for the calcineurin-dependent gating of the channel, we next examined whether the mutant channels retain the ability to respond to calcineurin activation. In WT TRESK channels expressed in *Xenopus* oocytes, an increase in intracellular Ca^2+^ produced by the calcium ionophore ionomycin generally results in strong calcineurin-dependent activation of TRESK [5,18]. We therefore determined the effect of ionomycin on WT TRESK and the identified variants. As expected, 0.5 μM ionomycin induced a large, robust, and reversible activation of WT TRESK (Figure 3A). By contrast, we found that ionomycin was much less effective in the activation of homomeric and heteromeric mutant channels (Figure 3A). Indeed, the absolute values of Ca^2+^-activated whole-cell currents were significantly smaller than that for the WT channel (Figure 3B; WT vs. all groups *p* < 0.001). Moreover, we found that the extent of activation (I_Iono_/I_Basal_ ratios) for both homomeric and heteromeric TRESK channels was much smaller compared to that observed for the WT channel (Figure 3C). However, due to the relatively small amplitude of the unstimulated I_Basal_ currents for the TRESK^Y163D^ and TRESK^S252L^ mutants, these values may be overestimated. The response of un-injected control oocytes to ionomycin was insignificant, as previously reported [13,19] (Figure 3A–C) and all groups showed an obvious difference in the ionomycin-induced current when compared to un-injected oocytes (Figure 3B; WT vs. un-injected *p* < 0.001; Y163D vs. un-injected *p* = 0.02; S252L and Y163D + S252L vs. un-injected *p* < 0.01). The relatively small ionomycin-induced current amplitude recorded from mutants, in particular from Y163D, could be responsible for the misleading reduced level of significance when compared with that of control un-injected oocytes.

Overall, these results indicate that although the identified disease-associated *KCNK18* variants do not alter the basal activity of TRESK, both markedly impair the ability of the channel to respond to calcineurin activation.

## 3. Discussion

TRESK background K^+^ channel contributes to the stabilization of the resting membrane potential of sensory neurons, particularly at the level of dorsal root and trigeminal ganglia [20,21]. Variants in the *KCNK18* gene have been frequently associated with migraine [9,11,13,22]. Interestingly, patients with ID, with or without migraine, have also been reported [12,13], suggesting a contribution of dysregulated TRESK function in NDDs.

In the frame of a large genome scan project focused on patients with NDDs [16], we identified a family of European origin with three siblings affected by ID. In the oldest case (patient 1), recurrent episodes of migraine were also reported. In this family, WES allowed the identification of a new compound heterozygosity for two missense variants (Tyr163Asp and Ser252Leu) in all affected siblings. The variant occurring on residue Ser252 caught our attention, as it has been largely demonstrated that this residue is fundamental for the regulation of TRESK, being subjected to reversible phosphorylation, regulating the activation of the channel [4,23,24]. In pharmacophore-based virtual screening to reveal novel inhibitors against migraine, it has also been suggested Ser252 is one of the binding sites for Ergotamine and Lasmiditan, two drugs commonly used for migraine treatment [25]. 

To functionally validate the pathogenic relevance of the identified amino acid substitutions on TRESK function, electrophysiological recordings using *Xenopus* oocytes were performed. Slightly larger whole-cell basal K^+^ current amplitudes were observed in oocytes expressing TRESK^S252L^, which could be attributed to a reduction in the phosphorylation state of the mutant channel compared to TRESK^WT^. Indeed, a similar effect has been reported when the serine in position 276 (TRESK^S276A^) in the rat clone was replaced with an alanine [26]. On the other hand, the substitution of serine 276 with a glutamate, mimicking the dephosphorylated state, resulted in a mutant with low basal activity and reduced responsiveness. Interestingly, TRESK^S276A^ also shows a reduced response to calcium signals. The S252 residue is part of the binding motif for 14-3-3, a family of ubiquitous adapter/regulatory proteins involved in the phosphorylation/dephosphorylation-dependent regulation of the channel. It has been shown that the 14-3-3γ isoform directly binds to the intracellular loop of TRESK and controls the kinetics of the calcium-dependent regulation of the channel [23].

Notably, 14-3-3γ plays a role in neuronal migration, morphology and brain development [27]. The gene encoding this protein (*YWHAG*) has been associated to developmental and epileptic encephalopathy (OMIM #717665), and autistic traits have been described in affected patients [28]. It is, therefore, intriguing to hypothesize that the S252L variant could cause an alteration in the interaction with 14-3-3γ, impairing its downstream activity and altering the same pathways involved in *YWHAG*-related disorders [28].

Here, we demonstrated that both the Y163D and S252L variants remarkably affect the TRESK response to intracellular Ca^2+^, which is able to cause calcineurin-dependent activation of the channel [5,18]. Specifically, while ionomycin induced a large and reversible activation of the WT channel, this activation was strongly reduced by the two variants, in both homomeric and heteromeric mutant channels. The exact location of both variants in the TRESK channel cannot be determined due to the lack of any 3D structural information on the regulatory domain of TRESK channels. However, in the linear sequence, Y163 is located nearby the calcineurin binding site. It could, therefore, be hypothesized that its substitution with aspartate (Y163D), a negatively charged residue, could disrupt calcineurin-channel interactions, and abolish current activation (Figure 3).

It is worth noting that the impairment of the channel activation after ionomycin induction (Figure 3) is not representative of the effect of the variants in the parents, as both the constructs were not expressed in combination with the WT one. With these experiments, we are, therefore, more likely mimicking a homozygous state of the variants instead of the heterozygous state observed in the parents. This is also in line with the higher impairment of the channel activation observed with Y163D and S252L alone compared to their combination.

The TRESK subfamily consists of only one member (*KCNK18*, K2P18.1) that has been cloned from the human spinal cord [29]. TRESK was shown to be expressed in rodent cerebrum, cerebellum, brain stem and, spinal cord [5]. Published evidence and data retrieved from the Allen Brain Atlas (www.portal.brain-map.org/ (accessed on 22 April 2021)) and GTEx Portal (www.gtexportal.org (accessed on 22 April 2021)) clearly show high expression of TRESK channels in human brain regions, for some of which the impairment has been associated with autism and ID [30]. How mutations in the TRESK channels identified so far [12,13] alter the functionality of these brain areas and result in autism and ID remains to be investigated. Whatever the final impact on the human brain of the mutations described here, which disrupt the calcineurin-dependent regulation of TRESK, may be, some insights could be gained by considering the effects of calcineurin inhibitors.

Indeed, Cyclosporin A and FK506 are inhibitors of the calcium/calmodulin-dependent protein phosphatase calcineurin, which abolish calcium-dependent TRESK activation. Neurological complications of Cyclosporin therapy are frequent and include dysarthria, headache, restlessness and seizures. Neurologic toxicity related to therapy with Tacrolimus (FK506), a macrolide calcineurin inhibitor, has also been reported, including restlessness, agitation, confusion, sleep problems, mental depression, dysarthria, headache and seizures. Intriguingly, dysarthria, restlessness, headache and seizures have been observed in our patients and it is tempting to speculate that these symptoms could be associated, at least in part, with Y163D- and S252L-induced disruption of calcineurin-dependent TRESK channel activation in the brain. Recently, it has been proposed that calcineurin regulation of two-pore potassium-leak channel activity (*Kcnk5b*) mediates activation of developmental gene transcription, which controls a broad number of developmental pathways [31]. Similarly to TRESK channels, inwardly rectifying K^+^ channels (Kir) control the resting potential of neurons and K^+^ homeostasis in the brain. Previous findings showed the importance of several Kir channel types in neurodevelopment, autism and ID [32,33,34,35,36,37,38]. Thus, growing evidence shows that K^+^ dependent bioelectric mechanisms can regulate not only the excitability of neurons, but also cell migration, proliferation, differentiation and gene transcription, thereby promoting coordinated developmental signaling. 

In summary, our study suggests that the *KCNK18* variants observed in our patients dramatically impair the ability of the TRESK channel to respond to calcineurin activation, leading to the alteration of a crucial function of the channel. Indeed, acetylcholine, glutamate or histamine activate native TRESK currents through G-protein coupled receptors [39], and this function would be nearly abolished by both Y163D and S252L variants. Collectively, our findings strengthen the notion that abnormal TRESK channel function in the central and peripheral nervous system could play crucial roles in neurologic and psychiatric diseases. Further studies on larger cohorts of patients and the characterization of transgenic animal models of the disease are, however, required to elucidate the link between migraine, autism, epilepsy and intellectual disability observed in some patients carrying TRESK variants.

## 4. Materials and Methods

### 4.1. Whole Exome Sequencing, Prioritization, and Variant Calling

DNA was extracted from total blood using the ReliaPrep Blood gDNA Miniprep kit (Promega, Madi-son, WY, USA) following the manufacturer’s protocol and quantified with a NanoDrop spectrophotometer (Thermo Fisher Scientific, Waltham, MA, USA). Array-CGH was performed using a 60 K whole-genome oligonucleotide microarray (Agilent Technologies, Santa Clara, CA, USA).

Patients were enrolled in the Autism Sequencing Consortium (ASC) project and their gDNA samples were sequenced at the Broad Institute on Illumina HiSeq sequencers, as previously described [16,40].

Whole exome sequencing (WES) raw data of the trio were processed and analyzed using an in-house implemented pipeline that has been previously described [41,42], which is based on the GATK Best Practices [43]. The UCSC GRCh37/hg19 version of genome assembly was used as a reference for reads alignment by means of the BWA-MEM [44] tool and the subsequent variant calling with HaplotypeCaller (GATK v3.7) [43]. We used SnpEff v.4.3 [45] and dbNSFP v.3.5 [46] tools for variants functional annotation, including Combined Annotation Dependent Depletion (CADD) v.1.3 [47], Mendelian Clinically Applicable Pathogenicity (M-CAP) v.1.0 [48] and Intervar v.0.1.6, for functional impact prediction [49]. Thereby, the analysis was narrowed to variants which affect coding sequences or splice site regions. Moreover, high-quality variants were filtered against public databases (dbSNP150 and gnomAD ver.2.0.1) so that only variants with unknown frequency or with MAF <0.1%, as well as variants occurring with frequency <1% in our population-matched database (~2000 exomes), were considered.

Further variant stratification in conformity with the American College of Medical Genetics and Genomics (ACMG) guideline [50], considering also the mode of inheritance and functional in-silico prediction of impact, allowed us to consider the final set of variants for possible associations with the phenotype. All variants are referred to GRCh37 annotation and to NM_181840.1.

Identified variants were confirmed by Sanger sequencing using standard conditions and the following primers: 5′-GGGAGATGGCAGAAGGTCTCTTTA and 5′-TTACTCCTCTCCATGGCTTGTGG; 5′-CAAACTTGGCACATGTCCTTCAC and 5′-GCATGACCCTGAAAGACAACACA.

Variants were analysed with the VarSome tool [51] as a starting point for further analysis. This allowed simultaneous evaluation of at least nine in silico predictors. Variants’ frequencies in the population were evaluated using the Genome Aggregation Database (GnomAD), browser version 2.1.1.

### 4.2. Constructs

Human TRESK was subcloned between the 5′ and 3′ UTR of the Xenopus β-globin gene in the oocyte expression vector, pFAW. The identified variants were introduced by site directed mutagenesis and confirmed by automated sequencing. mRNA for wild-type and mutant channels was synthesized using the T7 mMESSAGE mMACHINE kit (Ambion, Life technologies, Carlsbad, CA, USA) and mRNA concentrations were quantified by spectrophotometric analysis prior to injection.

### 4.3. Electrophysiological Recordings

*Xenopus laevis* oocytes collection and defolliculation were performed according to standard protocols which fall under the international standards of animal care, the Maltese Animal Welfare Act and the NIH Guide for the Care and Use of Laboratory Animals. TRESK wild-type or mutant mRNAs were microinjected into oocytes in equal quantities, unless otherwise stated. Forty-eight hours after injection, whole-cell currents were recorded at room temperature (20–22 °C) using the two-electrode voltage clamp method (Axoclamp-2B, Axon DIGIDATA 1550B, Axon Instruments, Foster City, CA, Axon). Basal K^+^ currents were measured in a low K^+^ extracellular solution (in mM: 95.4 NaCl, 2 KCl, 1.8 CaCl2, 5 HEPES pH 7.5 with NaOH). Recording electrodes (0.3÷1 MΩ) were backfilled with 3 M KCl. Currents were filtered at 100 Hz and digitized at 1 kHz for analysis. From a holding potential of −80 mV, 1-second-long voltage commands were applied from −120 mV to +60 mV, delivered in 20 mV increments. Currents were recorded using Clampex 10.7 software (Axon Instruments, Foster City, CA, USA). Ionomycin-induced TRESK currents were measured in high K^+^ solution (in mM: 17.4 NaCl, 80 KCl, 1.8 CaCl2, 5 HEPES, pH 7.5 with NaOH) at the end of 300 ms-long voltage steps from a holding potential of 0 mV to −100 mV. Ionomycin (free acid form) was made as a stock solution of 1 mM in DMSO and diluted in the high K^+^ solution to 0.5 μM on the day of the experiment. Data were analyzed with Clampfit 10.7 (Axon instruments, Foster City, CA, USA) and Igor programs. 

### 4.4. Statistics

Data are given as mean values ± standard error of the mean (SEM), where n represents the number of oocytes. Results were reproducible in at least 2–3 different batches of oocytes. Statistical significance was determined using a student’s *t*-test. When error bars are not shown, they are smaller than the size of the symbol.

### 4.5. Online Resources

www.gtexportal.org, accessed on 22April 2021

www.gnomad.broadinstitute.org, accessed on 22 April 2021

www.portal.brain-map.org, accessed on 22 April 2021

www.varsome.com, accessed on 22 April 2021.

## Figures and Tables

**Figure 1 ijms-22-06064-f001:**
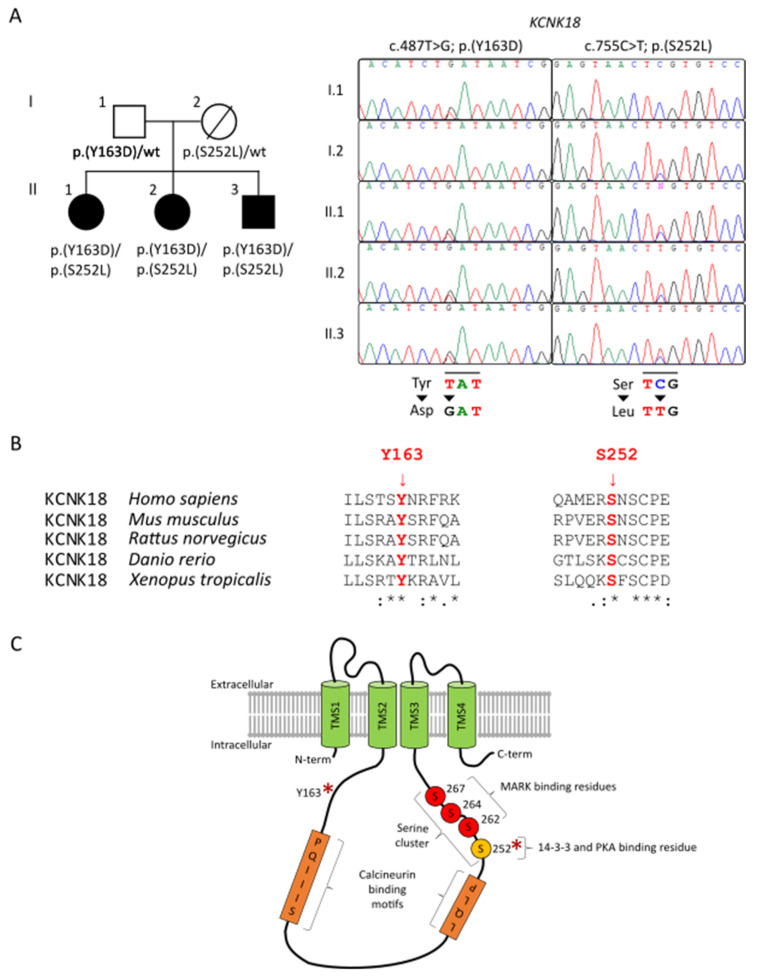
Pedigree, conservation, and graphical representation of the KCNK18 variants. (**A**). Pedigree of the family and Sanger sequencing chromatograms documenting co-segregation between compound heterozygosity for the two missense changes in KCNK18 and clinical traits transmitted in the family. (**B**). Multiple alignments of the amino acid stretches flanking the affected residues in TRESK orthologs showing conservation of both mutated residues (in red). Residues are referred to human KCNK18, NM_181840.1. (**C**). Schematic topology of the human K2P TRESK channel, showing location of Tyr163 (Y163) and Ser252 (S252) (red asterisks). Human TRESK is composed of 384 amino acid residues and shows the most characteristic features of K2P channels, including four transmembrane helices, two pore-forming domains, and intracellular N- and C-termini. A peculiar structural feature of TRESK (compared to other K2P channels) is the long intracellular loop between the second and third transmembrane segments (TMS), and the relatively short C-terminal tail [3]. In the image, representative domains are shown: PQIIIS and LQLP are calcineurin-binding motifs [5,17], while the serine cluster and Ser252 are indicated by the letter S and are, respectively, MARK kinases and 14-3-3 and PKA binding motifs.

**Figure 2 ijms-22-06064-f002:**
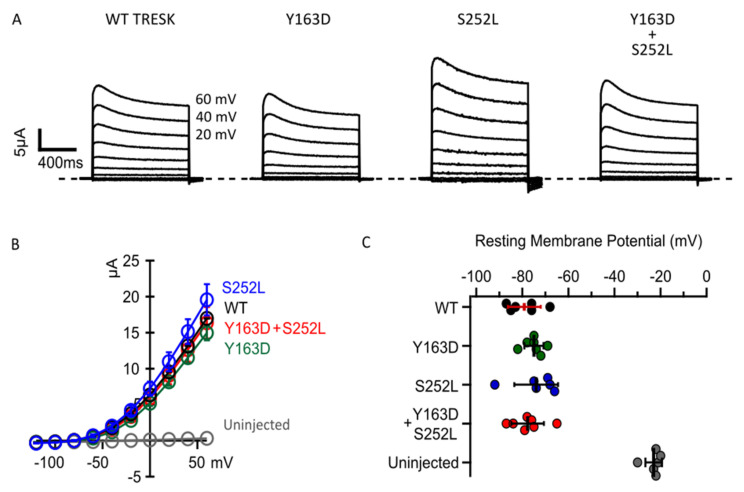
p.Tyr163Asp and p.Ser252Leu do not significantly affect the basal current and ability of TRESK channels to control the cell’s resting potential. (**A**). Representative families of current traces for the indicated channel types. Cell membrane potential was held at −80mV. Each family of currents was evoked by voltage steps from −120 mV to 60 mV, with 20 mV increments. The recordings were performed 48 h after the injection of 1 ng of mRNA for each channel type. (**B**). Average IV relationships calculated from experiments as in A for the labelled channel types (color coded). The data points are mean ± standard error (0.001; n = 12). (**C**). Resting membrane potentials from individual oocytes un-injected (grey dots), injected with 1 ng of WT (black dots) or Y163D (green dots) or S252L (blue dots) or 0.5 ng of each Y163D and S252L (red dots) mRNA and recorded 48 h after injection. Note that the expression of all channel types similarly shifts the resting potential toward K^+^ reversal potential. The dots in C represent single cell recordings. The data are mean ± standard deviation.

**Figure 3 ijms-22-06064-f003:**
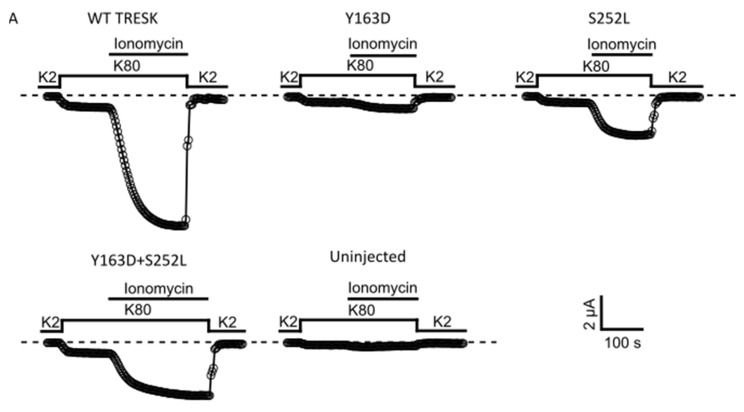
KCNK18 p.Tyr163Asp and p.Ser252Leu impair the ionomycin-induced activation of TRESK channels. Representative data points showing ionomycin activation of TRESK WT, Y163D, S252L and Y163D + S252L currents as well as endogenous currents from un-injected oocytes. (**A**). Currents were evoked by 300 ms long voltage steps from 0 mV to −100 mV from oocytes injected with 1 ng of mRNA of the corresponding and indicated homomeric channel type and 0.5 ng for each subunit for the heteromeric channel Y163D/S252L. The sampled data represent the average of 50 ms long period of the steady-state currents recorded at −100 mV. Ionomycin activates the WT channels but has little effect on the oocytes that express the mutant channels or those that are uninjected. Note that the current decay upon reapplication of a solution containing 2 mM K^+^ in the recording chamber is due to the reduced K^+^ concentration rather than ionomycin washout. (**B**). Bar graph showing the mean of ionomycin activated currents (*** *p* < 0.001). All the groups were statistically significant compared to the uninjected group (WT vs. uninjected *p* < 0.001; Y163D vs. uninjected *p* = 0.02; S252L and Y163D + S252L vs. uninjected *p* < 0.01; n = 12). (**C**). Each point represents the steady state ionomycin activated current divided by the steady-state basal current (I_Iono_/I_Basal_) recorded in the presence of 80 mM extracellular K^+^.

## Data Availability

The data that support the findings of this study are contained within this article and are available on request from the corresponding author. The WES data are not publicly available due to privacy or ethical restrictions.

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
