# Peer review of "KCNK18 Biallelic Variants Associated with Intellectual Disability and Neurodevelopmental Disorders Alter TRESK Channel Activity"

_ijms, 2021, doi:10.3390/ijms22116064_

Round 1

Reviewer 1 Report

  • Did the authors performed analysis on parents? No information is given.
  • The three siblings are not autistic ones, as stated in paragraph 2.1 (viceversa in the abstract!). However, they were enrolled in the ASC protocol. If they are not autistic patients, how is possible to correlate the association between the genetic variants and ASD?
  • Maybe a correlation with an autistic subject could help.
  • NDD could be more appropriate instead of ASD.

Author Response

  • Did the authors performed analysis on parents? No information is given.

The missense variants p.(Tyr163Asp) and p.(Ser252Leu) are in compound heterozygous state in all the three affected siblings, and they are inherited from the father and the mother, respectively, as showed in Figure 1A. These two variants have been functionally studied both alone and in combination, resembling the compound heterozygous state observed in the patients. 

Regarding the clinical information, the mother died of cancer some years ago; concerning the father, he did not report any noteworthy illnesses or symptom(s). We specified this point in the sentence: “The parents had no history of NDD or for notable diseases, except for a carcinoma in the mother.” (page 3, lines 109-110).

  • The three siblings are not autistic ones, as stated in paragraph 2.1 (viceversa in the abstract!). However, they were enrolled in the ASC protocol. If they are not autistic patients, how is possible to correlate the association between the genetic variants and ASD?

We thank the review for highlighting this inconsistency. The patients never received a formal diagnosis of autism spectrum disorders, meaning they were never evaluated using the Autism Diagnostic Observation Schedule (ADOS-2), nor the Communication and Symbolic Behavior Scale (BSBS).

On the other hand, they showed impaired social behaviour, with difficulties in the common relationship and always needed the support from an adult, leading the physicians to include their symptoms into the autism spectrum. We modified the paragraph to clarify this point.

  • Maybe a correlation with an autistic subject could help

We have not understood the request of the reviewer, and would be happy to know if he/she can expand his/her though, or if we already answered in the other covered points.

  • NDD could be more appropriate instead of ASD.
    We thanks the reviewer and agree with this consideration; we changed this concept in the title, using the term “neurodevelopmental disorder”.

Reviewer 2 Report

- Did the authors performed analysis on parents? No information is given.
- The three siblings are not autistic ones, as stated in paragraph 2.1 (viceversa in the abstract!). However, they were enrolled in the ASC protocol. If they are not autistic patients, how is possible to correlate the association between the genetic variants and ASD?
- Maybe a correlation with an autistic subject could help.
- NDD could be more appropriate instead of ASD.

Author Response

(The authors gave the same response as above.)

Reviewer 3 Report

The authors present an informative report of the variants within the TRESK channel. This manuscript describes novel missense variants in a family whose siblings are affected by ID and ASD and further provides functional data to support the role of these variants on channel properties  (e.g., basal current and regulation of resting potential) and calcineurin-dependent gating using oocytes. Overall, this manuscript is nicely written and easy to follow. The introduction and discussion provide ample background information and relevant supporting data to show the importance of this work. The addition of the functional experiments add additional evidence of the importance of investigating this channel and variants in NDDs. 

Minor comment: 

Figure 2B: There is a misspelling of "uninjected"

Author Response

The authors present an informative report of the variants within the TRESK channel. This manuscript describes novel missense variants in a family whose siblings are affected by ID and ASD and further provides functional data to support the role of these variants on channel properties  (e.g., basal current and regulation of resting potential) and calcineurin-dependent gating using oocytes. Overall, this manuscript is nicely written and easy to follow. The introduction and discussion provide ample background information and relevant supporting data to show the importance of this work. The addition of the functional experiments add additional evidence of the importance of investigating this channel and variants in NDDs. 

Minor comment: 

Figure 2B: There is a misspelling of "uninjected"

We thank the reviewer. The misspelling was edited and checked throughout the text.

Reviewer 4 Report

Pavinato et al reported two missense vairiants  of TRESK KCNK18 gene in a family characterized by three siblings affected by mild-to-moderate Intellectual Disability and autism spectrum disorder.

 Specifically etherozigous  Tyr163Asp/ Ser252Leu  resulted paternally/maternally inherited by all the affected siblings. The authors suggest that these variant may be associate with Intellectual disability. Both affected residues are located within a key region controlling TRESK function. Therefore they performed electrophysiological analysis in Xenopus oocytes injected with mRNA encoding human wild-type (WT) or the mutant TRESK channels and observed that disease-associated KCNK18 variants do not alter the basal activity of TRESK, but impair the ability of the  channel to respond to calcineurin activation.

 Though the study is very interesting an quite well written same concerns  raised regarding following points:

Figure 3B: the authors assess that “absolute values of Ca2+-activated whole-cell currents were significantly smaller than that for the WT channel” but no pvalue were reported nor in figure nor in the text.

It seems that  each  single variant may impair channel function  compared to  WT . This mean that the single presence of any of the two variants may  be associated to ID?

 If so did the authors correlate the presence of single variant in parents with their Intellectual and/or behavioural performance? 

Also environmental conditions like socio economic status as well as Environmental stress were reported to play a role in intellectual disabilities. How is the family context regarding these points? It would be useful to take in account these factors too to evaluate or to exclude any environmental role in the cognitive and behavioural deficits of these probands.

  A more in deep description of significant results and a better characterisation of the parents and family context  are needed

Minor revision

Check in the abstract Tyr162asp  was reported instead of tyr163asp

Author Response

Pavinato et al reported two missense vairiants  of TRESK KCNK18 gene in a family characterized by three siblings affected by mild-to-moderate Intellectual Disability and autism spectrum disorder.

Specifically heterozygous Tyr163Asp/ Ser252Leu resulted paternally/maternally inherited by all the affected siblings. The authors suggest that these variants may be associate with Intellectual disability. Both affected residues are located within a key region controlling TRESK function. Therefore, they performed electrophysiological analysis in Xenopus oocytes injected with mRNA encoding human wild-type (WT) or the mutant TRESK channels and observed that disease-associated KCNK18 variants do not alter the basal activity of TRESK but impair the ability of the channel to respond to calcineurin activation.

Though the study is very interesting an quite well written same concerns raised regarding following points:

  • Figure 3B: the authors assess that “absolute values of Ca2+-activated whole-cell currents were significantly smaller than that for the WT channel” but no p value was reported nor in figure nor in the text.

We thank the review for highlighting this inconsistency. We have now updated the information in Figure 3B caption.

  • It seems that each single variant may impair channel function  compared to  This mean that the single presence of any of the two variants may be associated to ID?

It is true that altered currents were observed both with the single variants and with their combination after ionomycin activation. However, it must be noticed, that is not possible to completely mimic the contribute of these variants in vivo for different reasons. First, the genetic background of the parents could be different from the one observed in the patients, and the effect of other variants or polymorphisms on other genes could not been excluded. Moreover, the experiments performed with the single Y163D and S252L variants does not mimic the heterozygous state observed in the parents (p.Tyr163Asp/wt and p.Ser252Leu/wt), as they are not co-expressed with the wt construct. Instead, they are more likely mimicking an homozygous state; this is also in line with the higher impact of these variants on the channel function compared to their impact when they are combined, representing the compound heterozygous state. We developed this topic in the Discussion section.

  • If so did the authors correlate the presence of single variant in parents with their Intellectual and/or behavioural performance?

No suspect of intellectual disability was observed in the parents at the first evaluation, nor in the father after during the last evaluation (mother was deceased). We already pointed this point with the sentence “The parents had no history of NDD or for notable diseases, except for a carcinoma in the mother.” (page 3, lines 109-110).

  • Also, environmental conditions like socio economic status as well as Environmental stress were reported to play a role in intellectual disabilities. How is the family context regarding these points? It would be useful to take in account these factors too to evaluate or to exclude any environmental role in the cognitive and behavioural deficits of these probands.

It is true that environmental conditions could have an impact on the outcome of some forms of intellectual disability. It is anyway to be noticed that the cases here presented a complex phenotype with delay in the developmental milestones, language impairment, and epileptic seizures. All these symptoms, together with the recurrence of the variants in all affected cases, support a strong genetic basis of this disorder. The socio-economic status of the family did not seem noteworthy; the presence of three affected kids and the death of the mother may obviously have caused a worsening of the familiar situation, but not the onset of an already established disease.
We have commented this topic in the Discussion section.

  • A more in deep description of significant results and a better characterisation of the parents and family context are needed

Updated characterization of the parents was possible only for the father, as the mother was deceased. His evaluation did not bring to light any worthwhile aspect and he was healthy.

Minor revision

  • Check in the abstract Tyr162asp was reported instead of tyr163asp

Thank you for noticing this error. We edited it.

Round 2

Reviewer 1 Report

Authors well answered to my comments.

Author Response

We thank the reviewer for his/her positive comment.

Reviewer 4 Report

The authors did assess most of the concerns and the manuscript has been largely improved.

I appreciated that they added p significant values in the figure legends but I would suggest to better describe these p value level of significancy  also in the result body of text.

Moreover when the signifcance level is higher than 0.01 it is perferable to report the two digit p value (i.e p= 0.0N instead of p<0.05)                 

Author Response

We added the requested details in the text.